# Prediction of Cavity Length Using an Interpretable Ensemble Learning Approach

**DOI:** 10.3390/ijerph20010702

**Published:** 2022-12-30

**Authors:** Ganggui Guo, Shanshan Li, Yakun Liu, Ze Cao, Yangyu Deng

**Affiliations:** 1School of Hydraulic Engineering, Faculty of Infrastructure Engineering, Dalian University of Technology, Dalian 116024, China; 2Conservancy and Hydropower Engineering, Xi’an University of Technology, Xian 710048, China

**Keywords:** cavity length, optimization algorithm, interpretable model, ensemble learning model

## Abstract

The cavity length, which is a vital index in aeration and corrosion reduction engineering, is affected by many factors and is challenging to calculate. In this study, 10-fold cross-validation was performed to select the optimal input configuration. Additionally, the hyperparameters of three ensemble learning models—random forest (RF), gradient boosting decision tree (GBDT), and extreme gradient boosting tree (XGBOOST)—were fine-tuned by the Bayesian optimization (BO) algorithm to improve the prediction accuracy and compare the five empirical methods. The XGBOOST method was observed to present the highest prediction accuracy. Further interpretability analysis carried out using the Sobol method demonstrated its ability to reasonably capture the varying relative significance of different input features under different flow conditions. The Sobol sensitivity analysis also observed two patterns of extracting information from the input features in ML models: (1) the main effect of individual features in ensemble learning and (2) the interactive effect between each feature in SVR. From the results, the models obtaining individual information both predict the cavity length more accurately than that using interactive information. Subsequently, the XGBOOST captures more correct information from features, which leads to the varied Sobol index in accordance with outside phenomena; meanwhile, the predicted results fit the experimental points best.

## 1. Introduction

Flow aeration using aerators is an inexpensive and effective technique for preventing cavitation erosion in spillways [1]. It sets a ramp in the chute to stir up the high-speed flow, followed by an aeration cavity that entrains air into the water through the air supply ducts. As a critical parameter in evaluating the air entrainment efficiency of a chute aerator [2,3], cavity length has been the focus of many researchers, and different empirical correlations have been proposed in the literature. Rutschmann [4] and Chanson [5] used the classical jet trajectory computation to calculate cavity length. Wu [3] further took the effect of flow depth and transverse fluctuating velocity into account to calculate the emergence angle, and the predicted cavity length presents a significantly better agreement with the experimental data than other correlations. Pfister [6] conducted dimensional analysis in the cavity length prediction and found that it is mainly affected by the aerator geometry of the chute bottom and deflector angle.

Though regression equations provide a reasonable estimation of the cavity length, their limited range of applicability and accuracy have also been reported in previous studies [7,8,9] due to their inability to capture the complex interactions between different input parameters. In recent years, machine learning (ML) models have gained popularity in modeling such problems. Initially stand-alone ML methods include artificial neural networks [10], SVR [11,12], adaptive neuro-fuzzy inference systems (ANFIS) [13,14], and multiple linear regression [15]. In the field of hydraulic engineering, most of the abovementioned algorithms are sole learning algorithms, whereas the ensemble learning techniques, which are more accurate, robust, and powerful [16,17], are applied to solve our problems. Qiu et al. [18] combined extreme gradient boosting (XGBOOST) with different optimization algorithms, such as the grey wolf optimization, whale optimization algorithm, and BO algorithm, to predict blast-induced ground vibration with 150 datasets and obtained a high prediction performance. Afan [19] conducted a study to predict the groundwater level using a combination of ensemble and deep learning models (ensemble DL). This study finds that ensemble DL is a more reliable tool compared to the individual DL model. Chen [20] estimated the wall shear stress for an ultra-high-pressure water-jet nozzle based on a hybrid BPNN model. Wu et al. [21] used XGBOOST to identify the water leakage zone and predict the leakage level, and their results confirmed that the prediction accuracy of the XGBOOST algorithm was better than that of the BPNN algorithm. However, the above-mentioned studies only focused on improving the accuracy of prediction, whereas no attempt has been made to further interpret the ML models.

In recent years, researchers started to focus on the interpretability of the ML models, which can bring significant improvements in the prediction accuracy and make models more credible [22,23,24,25]. For example, by calculating the contribution of each feature, the SHAP (Shapley Additive Explanations) method [26] uses the Shapley value to explain the prediction results. However, it only has the capability of extracting the effect of each individual feature in the prediction. For the purpose of revealing both the individual and interactive effects of the input parameters, the Sobol method [27] is proposed, which calculates the multiple sensitivity indices of the input features.

This study aims at developing ML models in the cavity length prediction of the aerator in the spillway. Experimental data from the State Key Laboratory of Hydraulic and Mountain River Engineering of Sichuan University and the Hydraulic Laboratory of Kunming University of Science and Technology are adopted, and the input parameters are selected based on an SVR test with 10-fold cross-validation. Then, the training is performed using three ensemble-learning models, i.e., RF, GBDT, and XGBOOST, respectively. The effect of the BO algorithm in optimization is also tested. Finally, the Sobol method was used to explore the relative significance of different input parameters in the cavity length prediction.

## 2. Machine Learning Models

### 2.1. Support Vector Regression

Vapnik [28] established the SVR model with the main goal of reducing the structural risk of complex problems by building a high-dimensional mapping relationship between input and output variables. Assuming that (*x*, *y*) is the observed data, where *x* is the input vector and *y* is the output of the observations, a linear relationship between the input and the output can be established as:(1)$$y^{\prime }=f(x)=\omega \phi (x)+b$$
where *y*′ is the predicted value of the model, *ϕ*(*x*) is the function that maps *x* from low to high dimensions, and *ω* and *b* represent the weight and bias of the model, respectively.

The reduction of the difference between the model outputs *y*′ and actual outputs *y* is achieved by minimizing the structural risk; the final nonlinear regression function is given as:(2)$$f(x)=\sum_{i=1}^{n}\left(\alpha_{i}^{*}-\alpha_{i}\right)K(x,x_{i})+b$$
where *α_i_* and *α*^*^*_i_* are the Lagrange multipliers, b is the bias of SVR, and *K*(*x, x_i_*) is the kernel function. The RBF radial basis function is adopted in this study.

### 2.2. Radom Forest (RF)

RF belongs to the bagging algorithm framework and has been adopted in many engineering problems [29]. RF utilizes two powerful algorithms, bootstrap aggregation and random subspace, to reduce the variance of the model and improve the generalizability.

In the construction of each RF regression tree, there are arbitrarily drawn subsets and features from the training dataset forming *n* bootstrap sets that are used to train *n_tree_* regression trees. The final output is the result of aggregating all regression trees:(3)$$y^{\prime }=\frac{1}{n_{tree}}\sum_{i=1}^{n_{tree}}{y_{i}}^{\prime }(x)$$
where *y*′ is the predicted value, and *n_tree_* is the number of decision trees.

Additionally, some of the data, named as out-of-bag data (OOB), are not included in the training. Instead, they are used to evaluate the performance of the trees, which can reduce the extra computational cost that comes with cross-validation [30].

### 2.3. Gradient Boosting Decision Tree (GBDT)

GBDT is an ensemble ML algorithm using the boosting framework. The establishment of each decision tree is not independent, and the establishment of the latter tree is based on the residual of the previous tree [31]. The residuals are reduced along the negative gradient direction in each iteration, and the final output is the weighted average of all the decision trees. The procedure of building the decision tree is as follows:(1)Initialize the iteration starting point *h_0_*(*x*).
(4)$$h_{0}(x)=\underset{f}{\arg\min}\sum_{i=1}^{n}L(y,y^{\prime })$$
where *h*_0_(*x*) is the initialized regression tree-based learner and *L*(*y*, *y*′) represents the error between the true value *y* and the predicted value *y*′ of the regression tree.

(2)The *m_th_* residual along the gradient direction is:

(5)$$r_{mi}=-\frac{\partial L(y_{i},h_{m-1}(x))}{\partial h_{m-1}(x_{i})},(i=1,2,\ldots ,n)$$where *r_mi_* represents the pseudo-residual, *h_m −_*
_1_(*x*) represents the prediction of the *m* − 1th tree, and *n* represents the number of samples.

(3)The establishment of the *m_th_* tree depends on the dataset *x* and *r_mi_*. Each sample has a prediction result *y*′ that is used to update the *m_th_* strong learner to obtain *h_m_* (*x_i_*).(4)After completing *m* iterations, the final strong learner *H*(*x*) is obtained.

### 2.4. Extreme Gradient Boosting

The XGBOOST algorithm, which can play a powerful role in gradient enhancement, was proposed based on the GBDT structure [32]. The main difference between XGBOOST and GBDT is that XGBOOST adds a regularization term to the loss function *L*(*y*, *y*′) to form the objective function *O*(*y*, *y*′):(6)$$O(y,y^{\prime })=L(y,y^{\prime })+R(f)+C$$
where *L*(*y*, *y*′) measures the difference between the prediction *y*′ and the target *y*, *R*(*f*) represents the traditional regularization term that is used to penalize the complexity of the model and helps to smoothen the final learned weights to avoid overfitting, and *C* is a constant.

Moreover, XGBOOST adopts a second-order Taylor series of the objective functions to optimize the objective quickly in a general setting:(7)$$O(y,y^{\prime })=\sum_{i=1}^{n}[g_{i}f(x_{i})+\frac{1}{2}h_{i}f^{2}(x_{i})]+\alpha T+\frac{1}{2}\eta {\left\|w\right\|}^{2}$$
where *g_i_* and *h_i_* denote the first and second derivative of the loss function, *α* is the weight of the min split loss, *T* indicates the number of leaves, *η* represents the weight of the regularization term, and *w* is the output of each leaf node.

### 2.5. Bayesian Optimization

BO can effectively handle a correlation whose mathematical expressions are unknown, and the calculation complexity is high [33]. The BO algorithm combines the probabilistic model with the acquisition function to find the minimum value of the goal function *f*(*x*). The probabilistic surrogate model adopts a Gaussian process that can quickly obtain the prior distribution of the function. The distribution of the function is typically determined by the mean *μ*(*x*) and variance *σ*(*x*).

Then, in each iteration (*t* = 1, 2,...*T*), the acquisition function uses *μ*(*x*) and *σ*(*x*) to select the point *x_t_* with the highest confidence in the *t^th^* iteration (*t* = 1, 2,...,*T*) and adds the point (*x_t_*, *f*(*x_t_*)) in the next iteration to approximate the function. The acquisition functions mainly include *a_PI_* [34], *a_EI_* [35], and *a_LCB_* [36]. The equations are:(8)$$a_{PI}=\Phi (\gamma (x)).\gamma (x)=\frac{f(x_{best})-\mu (x)}{\sigma (x)}$$
(9)$$a_{EI}=\sigma (x)(\gamma (x)\Phi (\gamma (x)))+\varphi (\gamma (x))$$
(10)$$a_{LCB}=\mu (x)-\kappa \sigma (x)$$
where *x_best_* is the best value point in the current iteration; *μ*(*x*) and *σ*(*x*) represent the mean and variance, respectively; Φ and *φ* are the cumulative distribution function and probability distribution function of the standard normal distribution, respectively; and *k* is the weight coefficient to balance *μ*(*x*) and *σ*(*x*).

The BO in this study adopts the *a_EI_* criterion and uses the cross-validation method to determine the combination of the hyperparameters. Their relative performance is evaluated using *R^2^*.

### 2.6. Model Evaluation Indices

To evaluate the accuracy of the prediction of the machine learning models, the coefficient of determination (R^2^), root mean squared error (RMSE), and mean absolute error (MAE) [37,38] are calculated, with their definitions being:(11)$$\mathrm{RMSE}=\sqrt{\frac{1}{N}\sum_{i=1}^{N}{\left(y_{i}^{*}-y_{i}\right)}^{2}}$$
(12)$$R^{2}=1-\frac{\sum_{i=1}^{N}{\left(y_{i}^{*}-y_{i}\right)}^{2}}{\sum_{i=1}^{N}{\left(y_{i}^{*}-\bar{y_{i}}\right)}^{2}}$$
(13)$$\mathrm{MAE}=\frac{1}{N}\sum_{i=1}^{N}\left(y_{i}^{*}-y_{i}\right)$$
where *N* represents the number of samples. *y_i_*, is the original experimental data, and *y_i_* represents its mathematical average. *y_i_^*^* is the result derived from the ML models.

### 2.7. Sobol Sensitivity Analysis

To acquire further understanding of the relative significance of different input parameters in the development of ML models, Sobol sensitivity analysis [27] is also carried out. The established model is assumed to be expressed as a function *Y* = *f*(*X*), where *X* = (*x*_1_, *x*_2_, …, *x_n_*) is the model input and *Y* is the output. The Sobol method can decompose the function *f*(*X*) into a sum of *2^n^* increasing terms:(14)$$Y=f(X)=f_{0}+\sum_{i=1}^{n}f_{i}(x_{i})+\sum_{1\leq i<j\leq n}^{n}f_{i,j}(x_{i},x_{j})+\cdots +f_{1,2,\cdots ,n}(x_{1},x_{2},\cdots ,x_{n})$$

Based on the variance decomposition method, the first-order index *S_i_*, second-order index *S_i,__j_*, and total effect index *ST_i_* are:(15)$$S_{i}=\frac{V_{i}}{V(Y)}$$
(16)$$S_{ij}=\frac{V_{ij}}{V(Y)}$$
(17)$$ST_{i}=1-\frac{V_{\sim i}}{V(Y)}$$
where *V*(*Y*) is the total variance of *Y*, *V_i_* is the variance of the *i^th^ X*, *V_ij_* is the variance of the interaction between the two variables *X_i_* and *X_j_*, and *V_~i_* is the average variance of all variables except for *i*. *S_i_* represents the contribution of variable *i*, also known as the main effect of *X_i_*; *S_ij_* represents the effect of the interaction of variables *X_i_* and *X_j_*; and *ST_i_* represents the combined effect of variable *x_i_* and its interaction with other variables.

To accurately calculate the sensitivity index, it is necessary to make reasonable sampling of the input data [39]. The optimal Latin hypercube design of Saletlli [40,41] was used. Because of the necessity of calculating the second-order sensitivity index, sampling must be performed *n*(2*m* + 2) times [40], where *n* and *m* are the number of samples and the number of input features, respectively. Choosing more data than the number of test samples can help in calculating the sensitivity index accurately. Therefore, in this study, *m* = 4 and *n* = 500 (>270 sets of data) are selected with 5000 uniformly distributed points sampled for calculating the Sobol indices.

### 2.8. Dataset and Dimensional Analysis

To develop the soft prediction models discussed in this study, we used 270 experimental data points from two universities: the State Key Laboratory of Hydraulic and Mountain River Engineering of Sichuan University and the Hydraulic Laboratory of Kunming University of Science and Technology. The experiments were performed in inclined rectangular open channels with channel widths of 30 cm and 10 cm, respectively. The experimental setup is illustrated in Figure 1. Air entrained by the ramp was supplied from the lateral duct to the cavity below the water nappe. A head tank is placed at the upstream side of the ramp that controls the flowrate. A Cartesian coordinate system is generated, with its origin located at point O. The *x*-axis has a direction parallel to the flume bottom centerline, whereas the *z*-axis is specified to be perpendicular to the bottom surface.

As a dominant factor that influences the air-entraining efficiency, the cavity length is influenced by the ramp height *s*, the ramp angle *α*, the chute bottom angle *φ*, the approaching flow velocity *v*, and the water depth *h.* The cavity length was measured from *x* = 0 to the impact point. Details of the two datasets are listed in Table 1. The sub-pressure is approximately the atmospheric pressure with the cavity sufficiently open to the atmosphere. Therefore, dimensional analysis was performed to obtain Equation (18) of the cavity length:(18)$$\frac{L}{h}=f(\alpha ,\varphi ,Fr=\sqrt{\frac{v^{2}}{gh}},\frac{s}{h})$$

Figure 2 shows the technical route of this study, which mainly includes three steps: (1) selection of the model input, (2) establishment of different ensemble learning models compared with traditional empirical regression, and (3) model interpretability analysis.

## 3. Results

As the selection of the input features and the training set ratio is critical in the development of the ML models, a pre-processing step that adopts SVR is carried out to analyze the effects of the feature selection and the size of the training dataset. Details of the cases for the test are listed in Table 2.

The statistical analysis of the error produced from different inputs and training dataset sizes is illustrated in Figure 3. According to the Table 2, input 1, with a training set ratio of 90%, presents the best performance among other conditions, with CV-R^2^ = 0.915, CV-MAE = 0.0742, and CV-RMSE = 0.0589. CV-R^2^ decreases and CV-RMSE and CV-MAE increase by only changing the ratio of the training set. CV-R^2^ decreased by 3.4%, and CV-RMSE and CV-MAE increased by 13.9% and 19.2%, respectively, when the proportion declined from 90% to 50%.

Compared with the training set ratio, the input features were a dominant factor influencing the predicted accuracy. The performance is only 0.782, 0.118, and 0.093 for input3 of the 90% training set ratio. Therefore, the ML model learns more information from features to accurately predict the cavity length.

To improve the prediction ability of ML models, this study uses the input combination of input1 and the 90% training set ratio.

### 3.1. Cavity Length Prediction Using Different Ensemble Learning Models

#### 3.1.1. Effect of BO Optimization

The hyperparameters are first determined by the BO method using 72 prior observation points in the Gaussian progress with the 10-fold CV technique, followed by the implementation of different ensemble learning models including RF, GBDT, and XGBOOST, as well as SVR. The derived hyperparameters and the time required for training are listed in Table 3.

As shown in Table 4, GBDT requires the least time for training among the different ensemble learning models, which is about 10% of the time spent by RF.

Deviations between the predicted results and the experimental data are showed in Figure 4. The ML models that implement BO optimization always present smaller errors compared with the cases without BO optimization. All ensemble learning models provide results with a better agreement with the experiment compared with the SVR. Among the different ensemble learning models adopted, XGBOOST-BO presents the best performance.

#### 3.1.2. Comparison with Empirical Correlations

An evaluation of the relative performance of ML models with respect to the traditional empirical correlations is also carried out. Five widely implemented empirical correlations proposed in the literature are selected; their expressions are listed in Table 5. The comparison of the results derived from these correlations with the results from XGBOOST-BO, which has the best performance among different ML models, is illustrated in Figure 5. The marker points with different colors represent the results derived from different models. The lines are linear fits that correspond to the results with the same color. Apparent deviation from the line of *y* = *x* exists for the correlations of Rutschmann, Chanson, Yang and Pfister, whereas the results of Wu present a similar agreement as XGBOOST-BO with *y* = *x*, which may result from the consideration of the transverse turbulent flow velocity *u*′ and the water depth in their correlation. To further evaluate the relative performance of the correlations, the R^2^ of their results is also listed in Table 5. The results of Wu present the highest value of R^2^, compared with the R^2^ = 0.996 of XGBOOST-BO; the superior accuracy of ML models is further demonstrated.

### 3.2. Model Interpretation Using the Sobol Technique

#### 3.2.1. Global Interpretations

In this study, Sobol analysis is carried out to evaluate the relative importance of different input parameters in the training process. The total sensitivity indices, which include the effect of the first and the second indices, are listed in Table 6. Focus on the SVR model first; *Fr* and *s*/*h* are the predominant input features in cavity length prediction. Then, the first- and second-order indices, which present the main and interactive effects of the input features in the ML models, are shown in Table 6 and Table 7. The effect of an individual feature is the least important, as the magnitude of *S*_i_ is much lower than that of *ST_i_*. As a result, the SVR output can be concluded to be predominantly affected by the interaction among the different input features.

As for the ensemble learning models, in contrast to SVR, it can be observed that individual features play important roles in the learning process. Among all the input features, *Fr* can be found to be the most important factor, as the *ST_Fr_* presents a higher magnitude than other input variables among all the models implemented. All second-order indices are close to 0, except for *S_Fr,s/h_* (0.14, 0.12, and 0.10), indicating that the interaction between two variables is less important in predicting the cavity length. Thus, information from an individual input feature dominates the prediction results in the ensemble learning model.

Overall, the above analysis implies that the SVR model relies more on the interactive effects between two or more features in the prediction, whereas the ensemble learning models prefer to extract information from an individual input parameter.

#### 3.2.2. Physical Interpretation of the Sobol Analysis Results

In physical conditions, the variation of *s*/*h* has a significant effect on the cavity length at a relatively low *Fr* due to the dominant role played by the gravity of the water jet. As *Fr* increases, the impact of subpressure and backwater becomes prominent, which significantly limits the cavity length [4,40,41]. Even a small increase in subpressure can drastically reduce the cavity length [1]. As the variation in the subpressure and backwater become more sensitive to the change in *φ* [42] at relatively high flow velocities, it can be expected that the main factor that influences the *s*/*h* becomes *φ* at a high *Fr*.

To further evaluate the interpretability of the different ML models against the physics under different flow conditions, the variation of *ST_i_* for *α*, *φ*, and *s*/*h* at different *Fr* values is plotted in Figure 6. At 2.46 < *Fr* < 5.5, all ML models show an *ST_i_* of *s*/*h* higher than 0.5, implying its dominant effect on the cavity length at a relatively low *Fr*. When *Fr* > 5.5, an abrupt fluctuation of *ST* with respect to *Fr* exists for the ensemble learning models, whereas its variation in SVR presents a relatively more smoothed style. The magnitude of *ST_α_* drops to 0 among all ensemble learning models, which corresponds to the fact that *α* becomes invariant in the experiments at *Fr* > 5.5. However, the S*T_α_* of SVR still presents a magnitude near 0.6 at *Fr* > 5.5, implying the inability of SVR to model the actual effect of different input features. As the *ST* of *s*/*h* for XGBOOST presents a relatively high magnitude at a low *Fr* and decreases dramatically as *Fr* increases, which has good correspondence with the actual effect of *s*/*h* in the formation of the cavity, and the variation of the *ST* of *φ* also complies with the change in the relative significance of *φ* in the experiments, the compliance with the physics demonstrates its stronger robustness compared with other ML models when facing such a problem.

#### 3.2.3. Evaluation of Model Accuracy at Different *Fr* values

As a main factor that influences the cavity length, the accuracy in capturing the dependence of *L*/*h* on the *Fr* of different ML models is also evaluated. A sampling space composed of 5000 data points is generated by Sobol analysis for different Froud numbers. The variation of the average of the ML-derived *L*/*h* among the 5000 samples with respect to *Fr* is plotted in Figure 7.

When *Fr* < 5.5, a slight positive correlation between *L*/*h* and *Fr* exists, and all the ML models present similar results. When *Fr* > 5.5, an apparent increase in *L*/*h* with the increase in *Fr* can be found. Under this condition, a significant underestimation of *L*/*h* exists for SVR, whereas the RF and GBDT tend to overpredict the *L*/*h*. The results of XGBOOST present the best agreement with the experimental data, which may result from its ability to capture the actual relative significance of different input features in the prediction of cavity length.

## 4. Conclusions

In this study, a high-precision ensemble learning model for cavity length prediction in aerators of spillways is established. Sobol sensitivity analysis is conducted to further explain the ML models based on the relationship between features and cavity length. The following conclusions can be drawn:

The optimal input combination and ratio is selected using the SVR model. The hyperparameters of the ensemble learning models are selected by combining 10-fold cross-validation with BO technology. The results show that the BO algorithm significantly improves the prediction performance. Among the different models implemented, the XGBOOST-BO model presents the highest test accuracy (R^2^ = 0.964, RMSE = 0.051, and MAE = 0.036). Additionally, the results imply that the ensemble learning model always outperforms the SVR model, even without the implementation of the BO optimization.

As for the comparison with the empirical correlations, the correlation proposed by Wu et al. presents the best agreement with the experimental data among all correlations taken into account, whereas its performance is still a bit worse than that of XGBOOST-BO, indicating the superiority of ML models in cavity length prediction when compared with the empirical correlations.

Finally, attempts to interpret the ML model are made by carrying out the Sobol analysis. The results imply that the SVR prefers to extract interactive information from two features, including *Fr* and *s*/*h*, while the ensemble learning models rely more on the individual effect among all input features. The Sobol analysis at different *Fr* values implies that all ML models correctly predicted the dominant effect of s/h on the cavity length at relatively low *Fr* values (*Fr* < 5.5), whereas only XGBOOST captured the diminished effect of *s*/*h* as *Fr* became higher than 5.5. The difference in the compliance with the physics is reflected in the accuracy of the cavity length prediction at different *Fr* values, where the results from XGBOOST present the best agreement with the experiments.

It has to be noted that the input data adopted in this study come from a relatively simple aerator composed of a ramp. More complicated aerators can also have offsets together with the ramp. Though it has wider applicability in hydraulic engineering, it brings more input features and is therefore more challenging in the development of the ML models, which will be the focus of the further investigations.

## Figures and Tables

**Figure 1 ijerph-20-00702-f001:**
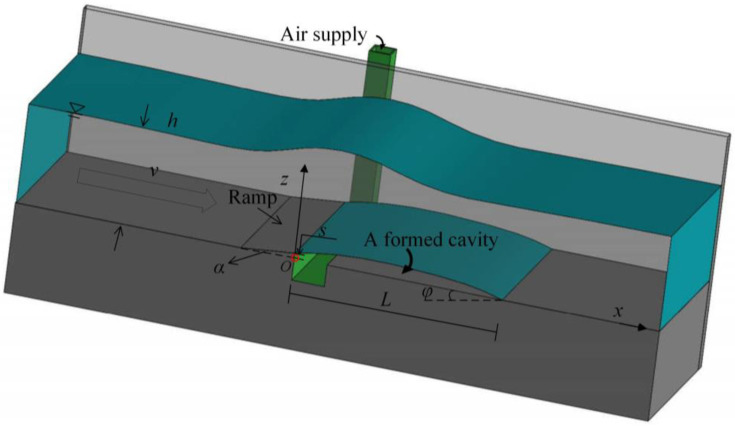
Schematic diagram of the spillway aerator and the formed cavity.

**Figure 2 ijerph-20-00702-f002:**
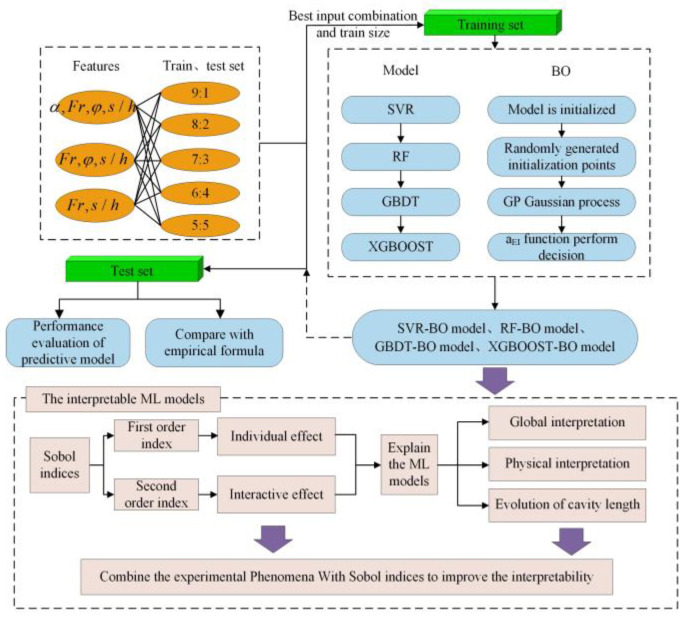
Construction of the ML models together with the interpreting analysis.

**Figure 3 ijerph-20-00702-f003:**
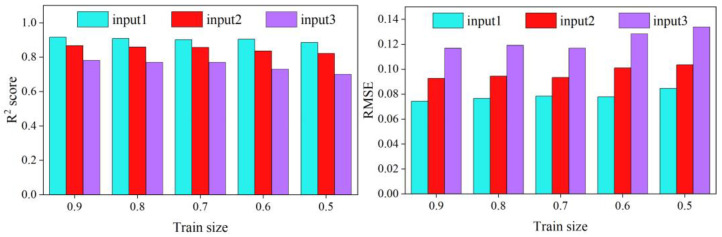
R^2^, RMSE, and MAE in cavity length prediction using the SVR model based on different input combinations.

**Figure 4 ijerph-20-00702-f004:**
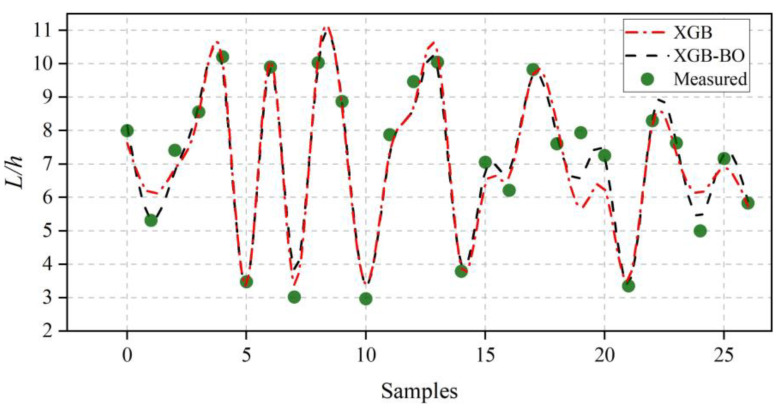
The prediction results of the optimal XGB-BO and XGB regressor on the test set.

**Figure 5 ijerph-20-00702-f005:**
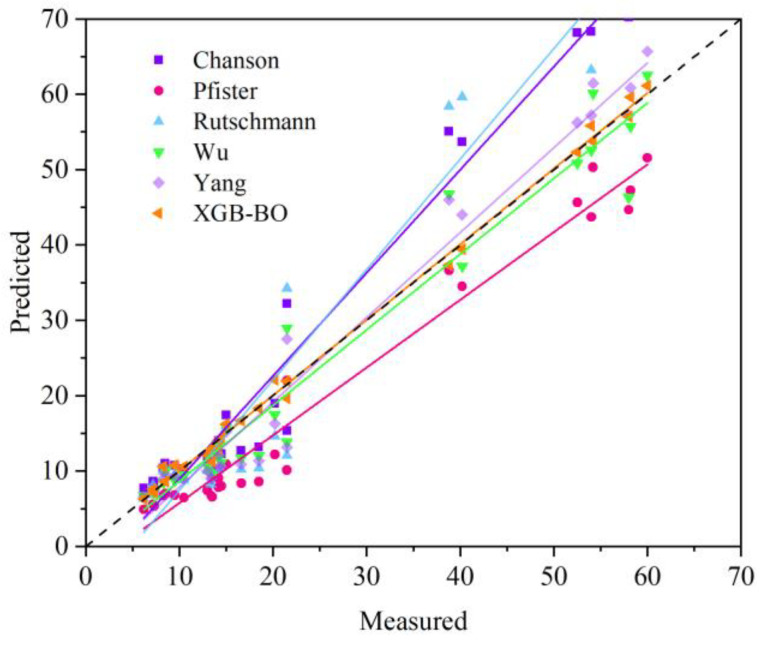
Cavity depth prediction derived from empirical correlations and XGBOOST-BO.

**Figure 6 ijerph-20-00702-f006:**
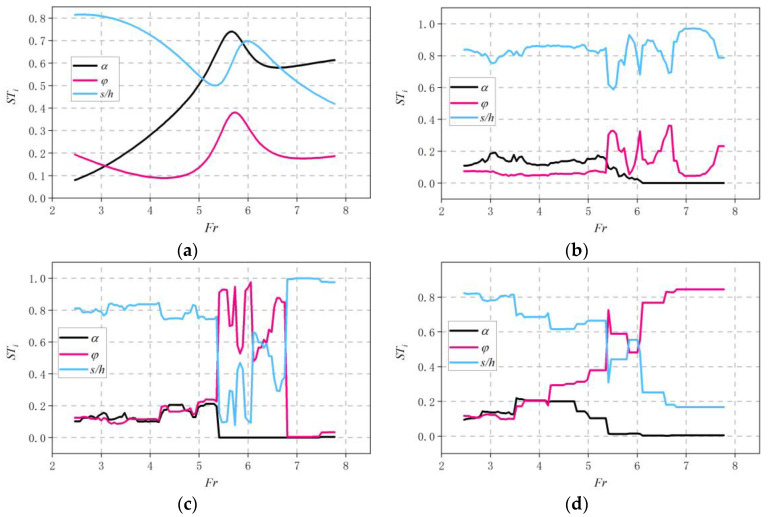
Evolution of the total Sobol′s indices. (**a**) SVR. (**b**) RF. (**c**) GBR. (**d**) XGB.

**Figure 7 ijerph-20-00702-f007:**
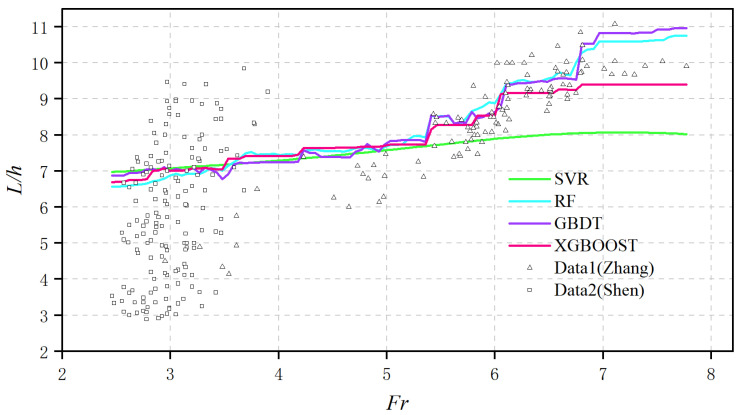
Prediction of *L*/*h* with respect to *Fr* using different ML models.

**Table 1 ijerph-20-00702-t001:** Parameter setting in the experiments.

	*α*	*h*/(cm)	*v*/(m/s)	*Fr*	*s*/(cm)	*φ*	*L*/(cm)	Samplers
Data1 (Zhang)	0.087 0.105 0.122	2.5~8.5	1.5~6.0	2.95~7.77	1.5 2.5 3.0 4.0	0.1	13.5~75.0	108
Data2 (Shen)	0.07 0.087 0.096	1.25~3.40	1.0~1.8	2.46~3.90	1.0 2.0 3.0	0.2 0.143 0.1	4.9~26.5	162

**Table 2 ijerph-20-00702-t002:** Input combinations of SVR.

Model	Input Configuration		Training Set Ratio (%)
	Name	Features	
SVR	input1	$\alpha ,Fr,\varphi ,\frac{s}{h}$	90 80 70 60 50
	input2	$Fr,\varphi ,\frac{s}{h}$	
	input3	$Fr,\frac{s}{h}$	

**Table 3 ijerph-20-00702-t003:** Hyperparameters optimization results.

Model	Hyperparameters			Time (s)
SVR-BO	*C* = 47.84	*γ* = 8.08	*ε* = 0.068	21.8
RF-BO	n_estimators = 28	max_depth = 50	——	305.1
GBDT-BO	n_estimators = 55	max_depth = 9	Learning_rate = 0.43	29.7
XGboost-BO	n_estimators = 47	max_depth = 3	Learning_rate = 0.29	145.7

**Table 4 ijerph-20-00702-t004:** Performance evaluation between ML models with and without BO optimization.

Model	Train			Test		
	R^2^	RMSE	MAE	R^2^	RMSE	MAE
SVR	0.931	0.067	0.06	0.921	0.081	0.060
SVR-BO	0.929	0.068	0.057	0.936	0.069	0.058
RF	0.989	0.0268	0.020	0.924	0.076	0.057
RF-BO	0.989	0.0264	0.020	0.947	0.063	0.050
GBDT	0.976	0.04	0.029	0.949	0.062	0.047
GBDT-BO	1.000	0	0	0.957	0.056	0.046
XGBOOST	0.999	0.00077	0.0016	0.921	0.077	0.055
XGBOOST-BO	0.964	0.015	0.038	0.964	0.051	0.036

**Table 5 ijerph-20-00702-t005:** Empirical correlations for cavity length prediction.

Reference	Equation	Parameters	R^2^
Rutschman, 1990 [4]	$\begin{array}{l} L=\frac{Fr^{2}\Theta d_{0}}{\cos\alpha }[1+\sqrt{1+\frac{2(t_{r}+t_{s})\cos\alpha }{\Theta^{2}Fr^{2}d}}](1\\ -0.4P_{N}{}^{0.5}) \end{array}$	$\Theta =\varphi \sqrt{\tanh(\frac{t_{r}}{d_{0}\varphi })}$, emergence angle	0.617
Chanson, 2010 [5]	$L=V_{0}T\cos\varphi +0.5gT^{2}\sin\alpha$	$\begin{array}{l} T=\frac{V_{0}\sin\varphi }{g(\cos\alpha +P_{N})}[1+\\ \sqrt{1+2(t_{r}+t_{s})\frac{g(\cos\alpha +P_{N})}{{\left(V_{0}\sin\varphi \right)}^{2}}}] \end{array}$	0.758
Yang, 2000 [42]	$L=V_{1}T\cos\varphi +0.5gT^{2}(\sin\alpha -0.00625Fr^{2})$	$\begin{array}{l} V_{1}=0.908V_{0}\\ \Theta =\varphi \sqrt{\tanh(\frac{t_{r}}{d_{0}\varphi })}\\ T=\frac{V_{0}\sin\Theta }{g(\cos\alpha +P_{N})}[1+\\ \sqrt{1+2(t_{r}+t_{s})\frac{g(\cos\alpha +P_{N})}{{\left(V_{0}\sin\Theta \right)}^{2}}}] \end{array}$	0.945
Wu, 2008 [3]	$L=V_{0}T\cos\Theta \prime +0.5gT^{2}\sin\alpha_{2}$	$\begin{array}{l} \Theta^{\prime }=0.48[\Theta +\alpha_{2}-\alpha_{1}]+\\ 0.52[\varphi +\alpha_{2}-\alpha_{1}-\arctan(\frac{u^{\prime }}{V_{0}})] \end{array}$	0.948
Pfister, 2010 [6]	$\left(\frac{L}{h})=0.77Fr{\left(1+\sin\varphi \right)}^{1.5}(\sqrt{\frac{s+t}{h}}+Fr\tan\alpha \right)$		0.868

**Table 6 ijerph-20-00702-t006:** First-order and total Sobol’s indices of Ml models with BO.

	*α*		*Fr*		*φ*		*s*/*h*	
	Total Index	First Index	Total Index	First Index	Total Index	First Index	Total Index	First Index
SVR-BO	0.235	0.0057	0.711	0.091	0.126	0.038	0.646	0.20
RF-BO	0.044	0.020	0.813	0.658	0.020	0.008	0.263	0.152
GBDT-BO	0.049	0.021	0.789	0.607	0.057	0.016	0.292	0.145
XGBOOST-BO	0.064	0.023	0.703	0.484	0.123	0.098	0.332	0.171

**Table 7 ijerph-20-00702-t007:** Second-order Sobol’s indices of different ML models.

	*α*, *Fr*	*α*, *φ*	*α*, *s*/*h*	*Fr*, *φ*	*Fr*, *s*/*h*	*φ*, *s*/*h*
SVR-BO	0.143	0.057	0.068	0.041	0.39	0.014
RF-BO	0.017	0.0	0.036	0.013	0.100	0.006
GBDT-BO	0.010	0.0	0.0	0.020	0.120	0.010
XGBOOST-BO	0.020	0.0	0.014	0.030	0.140	0.0

## Data Availability

Data is unavailable due to privacy or ethical restrictions.

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
