# Peer review of "Prediction of Cavity Length Using an Interpretable Ensemble Learning Approach"

_ijerph, 2022, doi:10.3390/ijerph20010702_

Round 1

Reviewer 1 Report

In this manuscript, the authors performed 10-fold cross-validation to select the optimal input configuration. The hyperparameters of three en-semble learning models — random forest (RF), gradient boosting decision tree (GBDT), and extreme gradient boosting tree (XGBOOST) — were fine-tuned by the Bayesian optimization (BO) algorithm to improve the prediction accuracy and compare the five empirical methods. The subject is very interesting. I am pleased to send you moderate comments. The results and theme of this paper is quite interesting. The layout is clear and easy to understand. Generally, this manuscript makes fair impression and my recommendation is that it merits publication in this Journal, after the following minor revision:

(1) The third paragraph of the introduction” In recent years, researchers start to put their focus on the interpretability of the ML models, which can bring significant improvement in the prediction accuracy [20].” A single reference is not sufficient here.

(2) What are the meanings of the symbols that appear in the paper?There should be a specific description. Such as in Eq.(2), What does the b mean? It is recommended that you make a symbolic description table.

(3) All the figures in this article are not clear enough, I hope they can be adjusted.

Reviewer 2 Report

This research provides a new kind of optimization algorithm and combination, and provides a new possibility and feasibility for the calculation and prediction of cavity length and related content.   

Formulas, figures and tables should be further improved by referring to the format specification of the journal, and the format of references should also conform to the specification.

In the part of analysis and conclusion, it is also necessary to emphasize and supplement the differences and connections with traditional algorithms and other methods.

Reviewer 3 Report

Detailed comments refer to the attachment.

Round 2

Reviewer 3 Report

Now this revision is acceptable.